# Different Approaches to Ergogenic, Pre-, and Probiotic Supplementation in Sports with Different Metabolism Characteristics: A Mini Review

**DOI:** 10.3390/nu15061541

**Published:** 2023-03-22

**Authors:** Jakub Wiącek, Joanna Karolkiewicz

**Affiliations:** Food and Nutrition Department, Poznan University of Physical Education, Królowej Jadwigi 27/39, 61-871 Poznań, Poland; wiacek@awf.poznan.pl

**Keywords:** sport supplements, ergogenic, gut microbiota, probiotic, prebiotic, dietary supplementation, athletes

## Abstract

Sport disciplines with different metabolic characteristics require different dietary approaches. Bodybuilders or sprinters (“anaerobic” athletes) need a high-protein diet (HPD) in order to activate muscle protein synthesis after exercise-induced muscle damage and use nitric oxide enhancers (such as citrulline and nitrates) to increase vasodilatation, whereas endurance athletes, such as runners or cyclists (“aerobic” athletes), prefer a high-carbohydrate diet (HCHD), which aims to restore the intramuscular glycogen, and supplements containing buffering agents (such as sodium bicarbonate and beta-alanine). In both cases, nutrient absorption, neurotransmitter and immune cell production and muscle recovery depend on gut bacteria and their metabolites. However, there is still insufficient data on the impact of an HPD or HCHD in addition to supplements on “anaerobic” and “aerobic” athletes’ gut microbiota and how this impact could be affected by nutritional interventions such as pre- and probiotic therapy. Additionally, little is known about the role of probiotics in the ergogenic effects of supplements. Based on the results of our previous research on an HPD in amateur bodybuilders and an HCHD in amateur cyclists, we reviewed human and animal studies on the effects of popular supplements on gut homeostasis and sport performance.

## 1. Introduction

Of seven different bacterial phyla, two seem to dominate the colon, namely *Firmicutes* and *Bacteroidetes* (together, constituting approximately 90%) [1]. Gut microbiota modifications occur mainly through diet, physical activity and drug intake changes, whereas only 8.8% of the diversity and abundance of bacteria are shaped by genes [2]. Some dietary choices, such as low-fiber and high-fat diets, may alter the gut microbiota within 24 h [3]. The disturbed ratio of *Bacteroides* to *Firmicutes* (B:F ratio) is an example of a microbial shift in obesity.

Exercise affects the gut microbiota and the intestinal environment. In general, a high level of physical activity is accompanied by an increase in gut microbiota diversity and health-promoting bacterial abundance (e.g., *Akkermansia muciniphila* and *Feacalibacterium prausnitzii*) [4]. However, there is high heterogeneity in the responsiveness of the human gut microbiota to the current lifestyle. It has been proposed that some people may be identified as good responders, while others are called non-responders because of a lack of microbial adaptations to lifestyle changes known to promote a healthy gut [5]. There are also indications that the microbiota accounts for ca. 58% of the variation in circulating metabolite levels in humans [6], which suggests that bacteria may be involved in nutrient and bioactive compound metabolism and absorption from not only food but also dietary supplements.

There is a growing body of knowledge on the impact of different diets (HPD, high-protein diet; HCHD, high-carbohydrate diet) and pre- and probiotics on athlete gut microbiota; however, nutrient intake level is not the only driver of the differences between the sport classification groups [7]. Little is known about the potential effects of specific supplements dedicated to different sport disciplines (e.g., citrulline in “anaerobic” exercise or sodium bicarbonate in “aerobic” exercise) on microbiota composition. Creatine and caffeine are two of the most effective sport supplements; however, the actions of these compounds are universal for sport disciplines of both types. In this review, besides the pre- and probiotics for athletes, typical anaerobic- and aerobic-type exercise supplements will be described, namely protein, citrulline, arginine, and nitrates for anaerobic disciplines and carbohydrates, sodium bicarbonate, and beta-alanine for aerobic disciplines.

The modification of athletes’ microbiota for performance enhancement remains under investigation. Clinical strategies for gut disorders may be used to enhance recovery after exercise and during increased training load season. Intestinal bacteria are known to affect sleep, appetite, mood, pain and cognition [8,9,10,11,12]. There are multiway connections between the gut microbiota and the brain, as well as peripheral organs, such as the lungs, skeletal muscles, liver and skin [13,14,15,16]. Mechanisms underlying many regulatory functions of the microbiota are neuroimmonologically based, as gut bacteria that are known to promote health influence production and modulate the biological activities of immune cells and neurotransmitters [17].

## 2. Aerobic Exercise—Diet and Supplements

Regular exercise modulates the gut microbiota, but it was found that endurance training affects the intestinal microbiota in a specific way [18]. Long-distance running or cycling increases abdominal organ ischemia, which may have detrimental effects on the intestinal epithelium [19]. Along with local ischemia or hydration status changes, dietary recommendations for endurance athletes are an additional potentially harmful factor that could affect the gut microbiota. It is recommended that runners or cyclists consume a very high-carbohydrate diet (>45% of calories and >6 g of carbohydrates (CHO) per kg of body weight) to maintain muscular glycogen stores and sustain energy levels during long-lasting training or competition [20]. It is also recommended for athletes in these disciplines to avoid excessive fiber consumption, as it may slow down digestion and cause gastrointestinal distress during exercise. These two recommendations make the diet for endurance athletes similar to a Western diet, which is known to promote weight gain and metabolic disorders.

However, high-carbohydrate, so-called agrarian diets promote microbial biodiversity and richness. It is hypothesized that this effect is mostly mediated by fiber and resistant starch consumption. Diets based on grains, fruits, and vegetables may decrease *Bacteroides* abundance, while increasing probiotic strains of *Bifidobacteria* [21]. While cyclists, runners, or swimmers avoid excess fiber, it could be a risk factor for a healthy gut. In our previous study, we did not observe any significant differences in *Bacteroides* spp., *Bifidobacterium* spp., *Akkermansia muciniphila*, and *Feacalibacterium prausnitzii* abundance between amateur cyclists consuming a high-carbohydrate diet and controls with a more sedentary lifestyle consuming a diet containing more protein and fat [22]. Although the carbohydrate consumption of study participants was high (mean of 4.48 g/kg b.w. (body weight) in precompetition season vs. 5.18 g/kg b.w. in competition season; *p* < 0.05), it was not as high as in professional athletes (6–10 g/kg b.w.). The cyclists in the study also consumed appropriate amounts of fiber (approximately 26.8 g daily). These observations confirm the results of a study conducted by other authors who used a different method of bacterial genome sequencing and collected more samples. They found higher abundances of *Bacteroides* and *Blautia* in marathon and half-marathon runners and competitive cyclists than in controls, higher *Veillonella* in runners than in controls, and higher *Prevotella* in cyclists with a high training load than in cyclists with a low training load [23]. The same effect for *Prevotella* was observed in marathon runners and cross-country skiers (as compared to sedentary controls) but not for *Bacteroidetes* [24].

Glucose (dextrose) and maltodextrin (hydrolyzed starch) are among the most popular supplements for endurance athletes. Using CHO-based sports drinks at doses up to 90 g per training sessions (or 1.2 g/kg b.w.) enhances glycogen stores and maintains hydration and speeds up the recovery process [25]. However, high sugar consumption is known to exert negative effects on the gut mucosa and epithelium and, therefore, on the microbiota. This, in turn, may promote low-grade inflammation related to cardiovascular disease and other disorders [26]. The authors of these findings called maltodextrin “the modern stressor of the intestinal environment”. Gut dysbiosis a type 2 diabetes mediator, and high sugar intake was observed to decrease *Bacteroidetes* and increase *Proteobacteria* [27]. Drinks with glucose and/or maltodextrin, which contain 10–15% of daily carbohydrates, may significantly increase total carbohydrate intake and potentially interfere with both the gut protective barrier and the microbiota. Carbohydrate mouth rinse could be considered to maintain hydration during exercise, but it does not allow for larger doses of carbohydrates [28].

Athletes use sodium bicarbonate (NaHCO_3_) to buffer excessive hydrogen ion accumulation in muscles during exercise. Some data suggest that it enhances endurance performance, but the outcomes of meta-analyses have yielded conflicting results [29]. To date, there have been no studies on the potential impact of continuous sodium bicarbonate intake on the gut microbiota composition and metabolome of athletes. However, it was found that NaHCO_3_ at doses over 0.2 g/kg b.w. can acutely trigger adverse gastrointestinal symptoms [30]. These symptoms include diarrhea, which not only leads to dehydration but also microbiota disturbances. In one study, researchers assessed gut microbiota changes in patients with liver steatosis using water with standardized sodium bicarbonate, calcium, magnesium, and sulfate contents. They found a decrease in the abundance of *Blautia* strains and an increase in *Subdoligranulum*, both of which have potential probiotic properties [31]. In patients with type 2 diabetes, it was found that drinking bicarbonate-enriched water is associated with changes in metabolites related to carbohydrate breakdown and an increase in *Dehalobacteriaceae* bacteria strains [32]. It is unknown whether NaHCO_3_ has any effect on the gut microbiota of athletes.

Beta-alanine is one of the compounds that make up carnosine in the body and effectively increases its level when consumed with diet [33]. Carnosine is a peptide that is mostly stored in muscle tissues. Among its most important functions are to neutralize reactive oxygen species, reduce glycation and chelate metal ions. It also blocks the accumulation of hydrogen ions in skeletal muscles during high-intensity physical activity, which is why athletes often use it [34]. In ergogenic doses, that is, 4–6 g daily divided into 4–5 portions, beta-alanine is considered well-tolerated [35]. No human studies have considered the potential effects of beta-alanine on the gut microbiota.

## 3. Anaerobic Exercise—Diet and Supplements

Resistance exercise (especially in the eccentric phase of a given exercise) and sprints (especially during downhill running) are known to cause muscle damage. To recover from the most intense strength and speed training, athletes consume a high-protein diet with special focus on branched-chain amino acids (BCAAs; mainly leucine, isoleucine, and valine) from sources such as whey protein, eggs and meat, which stimulates muscle protein synthesis through the mTOR pathway.

There is a limit to the effective use of proteins, and excessive consumption of protein sources may negatively affect the gut microbiota. The generally accepted limit is set at around 1.6–2.2 g of protein per kg of b.w. divided across 4–5 meals [36], with a maximum of 0.4–0.55 g/kg b.w. (4 meals) or 0.32–0.44 g/kg b.w. (5 meals). The amount of protein that exceeds dietary recommendations may turn into toxic metabolites (e.g., ammonia and amines) through proteolytic fermentation. The gut microbiota is a key regulator of this process [37]. However, the relationship between the protein consumption level, protein sources, processing methods, physical activity type, and gut microbiota remains unclear [38]. In our recent publication, we compared the gut microbiota composition of amateur bodybuilders on a HPD and sedentary controls on a diet containing more fat (mean calories from protein: 33.6% vs. 22%, respectively, *p* < 0.05; mean calories from fat: 27.6% vs. 36.4%, respectively, *p* < 0.05). We observed no significant differences in the colony-forming unit counts of selected intestinal bacteria (e.g., *Bacteroides* spp., *Bifidobacterium* spp., *Akkermansia muciniphila*, and *Feacalibacterium prausnitzii*) [39].

Different sources of dietary protein (animals, plants, mushrooms and yeasts) may have different impacts on the gut microbiota due to different fiber and antioxidant contents. However, protein supplements such as concentrates and isolates have most of the fat, carbohydrates, and fiber removed and are therefore easily digestible. Among athletes, whey products seem to be the most popular [40]. A systematic review of eight randomized controlled trials showed that, contrary to yogurt or kefir, whey and casein isolates (from milk) do not significantly affect the gut microbiota composition in healthy people [41]. In a randomized clinical trial, it was found that protein supplementation during caloric restriction leads to greater visceral fat mass reduction and increased microbial diversity, especially in participants with low baseline diversity, as compared to a diet without additional protein [42]. In infants (1–3 years old), whey protein hydrolysate induced an increase in the production of probiotic bacteria counts and metabolites (short-chain fatty acids; SCFAs), which suggests prebiotic functions of hydrolyzed protein [43]. However, in a pilot trial of the impact of protein supplements on athlete gut microbiota, a team of researchers found a decrease in probiotic strains of *Blautia* and *Bifidobacterium* (*Bifidobacterium longum*) and an increase in *Bacteroidetes* [44]. The authors concluded that long-term protein supplementation may have detrimental effects on the gut microbiota; however, this study was conducted on a small group (protein supplementation, n = 12; control, n = 12) of endurance rather than resistance athletes. In another randomized, double-blind trial examining the effects of multicompound products based on whey protein on sleep quality and the gut microbiota of people with sleep problems, researchers observed an increase in *Bifidobacterium* abundance [45]. However, it is not clear whether this effect was achieved through whey protein or galacto-oligosaccharides, which are known for prebiotic properties and were a part of the tested product. The impact of soy protein and peptides on gut microbiota seems to be more unequivocal. In a mini review based on both animal and human studies, the authors found that soy derivatives stimulate the growth of microbial diversity, especially bacteria with probiotic properties [46]. There is evidence that soy peptides stimulate *Lactobacilli* and *Bifidobacteria* and simultaneously decrease *Bacteroidetes*, which is why athletes should consider mixing their protein sources in the diet.

Bodybuilders specifically value supplements such as citrulline and arginine because these amino acids promote vasodilation through increased nitric oxide (NO) production. This effect (so-called muscle pump) increases the transport of oxygen to working muscles. Citrulline is an amino acid derivative whose metabolism is related to the protein amino acid arginine [47]. When consumed in the diet, citrulline is broken down into arginine molecules. In turn, this amino acid is involved in the synthesis of nitric oxide in the endothelial cells of blood vessels. It delays the onset of fatigue during strength training and reduces muscle soreness on the first day after intense exercise. Another mechanism of action of citrulline or citrulline malate is the excretion of excess ammonia, which is formed during muscle activity and contributes to fatigue [48]. In addition to ammonia clearance, citrulline can improve gut homeostasis. In a double-blind, crossover study of 10 healthy men, citrulline supplementation prior to exercise attenuated splanchnic hypoperfusion, thereby protecting the mucosa from exercise-induced damage [49]. It has been proposed that this effect is mediated by increased arginine bioavailability. Citrulline and arginine participate in the urea (ornithine) cycle. Moreover, citrulline is a diagnostic tool for assessing short bowel function, as it is produced mostly in the gut [50]. While citrulline and arginine are recommended in both types of sport disciplines (aerobic and anaerobic), athletes such as bodybuilders and weightlifters tend to use much higher doses than runners and cyclists. In aerobic disciplines, athletes should consume approximately 1.5–2.0 g of arginine per day, while athletes in anaerobic disciplines may take more advantage of doses up to 10–12 g per day [51]. There is evidence that arginine, similar to glutamine, contributes to SCFA levels, thereby reducing the ratio of *Firmicutes* to *Bacteroidetes* [52]. Owing to its alkalizing properties, arginine is used as a prebiotic agent in dental care [53].

Citrulline and arginine, as well as dietary nitrates, are consumed by athletes for the same reasons. While the abovementioned amino acids increase NO indirectly, dietary nitrates (for example, from beetroot and rocket) do so directly. Nitrate pathways are mediated by microbial communities in the gut and provide a respiratory substrate [54]. Nitrate supplementation is one of the most effective methods to enhance exercise performance. Nitrate reduction begins in the mouth and is induced by specific bacteria [55]. While its properties in the muscular system, gastrointestinal tract, and oral microbiota are well known, its potential impact on the gut microbiota of athletes of different sports remains unknown. In one study conducted on human fecal samples, it was found that NO may decrease health-promoting *Faecalibacterium prausnitzii* biomass [56]. In this experiment, researchers used an in vitro fermentation model to mimic the natural gut environment.

## 4. Prebiotics for Athletes

Prebiotics are a group of substances resistant to enzymes present in the human digestive tract and capable of stimulating the growth of health-promoting microorganisms. These substances improve the colonization of the host organism, which is a desirable phenomenon from the point of view of the functioning of many areas of the entire body.

Pectins (mainly from fruit) are non-digestible oligosaccharides that delay gastric emptying and lower blood glucose [57]. In a recent review, pectin fermentation was found to promote the abundance of *Bacteroides* and *Faecalibacterium prausnitzii* [58]. Owing to their antihyperglycemic and prebiotic properties, pectins should be considered a basic element of carbohydrate products for endurance athletes. Additionally, there is growing interest in the impact of sodium alginate on glycemic control. However, a meta-analysis found no ergogenic effects of drinks containing carbohydrates and sodium alginate [59]. Interestingly, in a study comparing the effects of carbohydrate drinks and pectin–alginate-enriched carbohydrate drinks on gut barrier status of athletes training in a hot–humid environment, researchers found no significant differences. Both drinks protected the intestines better than water [60].

Inulin (mainly from chicory) is another non-digestible carbohydrate that acts as a prebiotic. In a population of adults at risk of type 2 diabetes, inulin supplementation (10 g per day for 6 weeks) led to a reduction in homeostatic model assessment insulin resistance and an increase in *Bifidobacteria* [61]. Fructo-oligosaccharide-enriched inulin increased the abundance of *Bifidobacterium uniformis* in adults implementing high-intensity interval training [62]. Fructo-oligosaccharides, a group of carbohydrate derivatives similar to inulin, increase the number of *Bifidobacterium* species in the gut. The efficacy of doses up to 15 g/day for 4 weeks was confirmed in a recent systematic review and meta-analysis of human studies [63].

Another type of prebiotic that could be helpful for athletes is beta-glucans (i.e., from mushrooms and oats), which may promote *Lactobacilli* and *Bifidobacteria* abundance and elevate the *Firmicutes*/*Bacteroidetes* ratio [64]. There are similarities in the prebiotic properties of inulin and beta-glucans [65]. Surprisingly, beta-glucan supplementation at doses of 2 g/day for 4 weeks was found to increase athletes’ grip strength [66]. Improvements in VO_2_max and 1 min double rocking jumps were also reported in this study. In healthy people exercising in the heat, beta-glucan (from yeast) was found to decrease inflammatory markers levels, which may preserve intestinal mucosa and microbiota in prolonged exhaustive activities [67].

## 5. Probiotics for Athletes

Probiotics are live microorganisms that have a mutualistic relationship with human cells when they are delivered to the gastrointestinal tract (supplements, fermented vegetables and dairy). Inhabiting the intestines, they produce protective compounds that strengthen the physical barrier between the lumen of the digestive tract and the bloodstream, as well as the microbiological barrier, by secreting compounds that inhibit other microorganisms [68]. By influencing the “lining” of the intestines, probiotic bacteria facilitate the absorption of electrolytes, controlling the state of hydration, and also improve the breakdown of proteins, fats and carbohydrates, modifying the nutritional state [69]. Much attention is also paid to the fact that bacteria produce vitamins, especially B vitamins, and enhance absorption of iron and calcium [70,71].

To date, there has been one sport-specific systematic review and meta-analysis of the effects of multistrain probiotic supplementation on the exercise capacity of endurance athletes. The authors found that probiotics increased the time to exhaustion, specifically when single-strain (e.g., *L. plantarum*, *L.casei*, and *B.longum*) probiotics were administered at doses over 3 × 10^9^ for less than 4 weeks [72]. Much less is known about the probiotic effects on muscle recovery in athletes with anaerobic metabolism predominance. According to a previous review, there is potential for gut microbiota modulation in the prevention of sarcopenia, but the overall data are limited [73].

The focus was on the probiotic effects on the immune systems of athletes and the number of training days missed because of respiratory tract infections. In one systematic review, a group of researchers concluded that probiotic supplementation resulted in a decrease in the risk of developing infections and symptom severity [74]. Modulation of the inflammatory cytokine profile has been proposed as the main mechanism underlying the immunomodulatory effects of probiotics in athletes [75].

## 6. Ergogenics and Gut Microbiota—Animal Studies

Both germ-free and rodent models with antibiotic treatments are often used to determine potential modulators of host gut microbiota. Germ-free animals are housed in a sterile environment, which allows for detection of non-environmental factors affecting intestinal bacteria [76]. In a recent experiment comparing the effects of endurance and resistance exercise on murine (C57BL6N mice) gut microbiota, endurance training was found to promote higher bacterial diversity. Four weeks of different training programs led to higher relative abundance of *Desulfovibrio* species in endurance exercise and *Clostridium* sp. (namely *C. cocleatum*) in resistance exercise [77]. These results support hypotheses on the different effects of different exercise characteristics on the intestinal microbiome. However, ergogenic supplements and their interactions with the gut microbiota in trained rodents were not studied extensively.

Diets with added maltodextrins induced intestinal inflammation in mice. This effect was caused by endoplasmic reticulum stress in the epithelium, with mucus depletion as a consequence [78]. Beta-alanine supplementation was not studied in the animal microbiome experiments. However, in mice receiving antibiotic treatment, researchers found a decrease in *Bacteroidaceae* and increase in *Prevotellaceae* and *Rikenellaceae*, along with changes in the metabolism of beta-alanine [79]. In the ischemia–reperfusion model of intestinal injury in rats, beta-alanine was found to attenuate tissue damage through decreased macrophage accumulation [80]. Sodium bicarbonate in swimming rats was found to prevent gastric retention and acid-based changes caused by exercise [81]. This could help sustain hydration status during exercise and avoid diarrhea or vomiting.

Whey protein isolate, in comparison with casein in C57BL/6J mice on a high-fat diet, increased *Lactobacillus murinus* and decreased parameters related to obesity. However, this effect was seen in younger but not older mice (5 vs. 10 weeks old) [82]. In another study, whey protein reduced weight gain in young mice but did not affect the microbiota composition significantly [83]. Interestingly, health benefits of whey were not seen in mice with microbiota depleted through antibiotics [84]. In obese animals, whey proteins promote an increase in *Bifidobacteria* abundance [85]. In addition, cheese whey protein has a protective potential in mild experimental colitis, as it increases *Lactobacilli* and *Bifidobacteria* counts and mucin production [86]. Citrulline supplementation in rats after small intestine resection (80%) led to nitrogen balance preservation and an increase in the arginine level relative to arginine alone. However, in this study, gut microbiota composition was not tested. [87]. Fourteen days of L-arginine supplementation in mice led to a shift in the *Firmicutes–Bacteroidetes* ratio, increasing *Bacteroidetes* counts. This change was associated with the regulation of innate immune signaling [88]. Arginine may also protect from intestinal integrity disruption and bacterial translocation, as observed in mice with intestinal obstruction [89]. Nitrates from diet were not studied in the context of the gut microbiota.

## 7. Prebiotics, Probiotics and Gut Microbiota—Animal Studies

Supplementation of inulin in male wild-type Groningen rats for 2 weeks increased *Bacteroidetes* and decreased *Firmicutes* abundances, along with increased acetate and succinate production [90]. In hyperuricemia mice, inulin enrichment of the diet led to a decrease in toxin levels and an increase in health-promoting *Akkermansia* bacteria, as well as SCFAs [91]. In a longer period of time, pectins from different food (beet, citrus, and soy) were found to increase *Firmicutes* and *Lactobacillus* and decrease *Bacteroidetes* in male Wistar rats (7 weeks of supplementation), and this shift was found to increase butyrate and propionate production [92]. Beta-glucans reversed gut barrier dysfunction in obese mice fed with a high-fat diet. This phenomenon was accompanied by regulation of *Bacteroidetes* and *Proteobacteria* levels, as well as cognitive changes [93]. Beta-glucans were also found to promote *Blautia* and *Alistipes* and inhibit *Proteus* and *Lachnospiraceae* and to be beneficial in the ulcerative colitis mouse model [94].

The list of probiotics studied by scientists is constantly expanding, as molecular techniques for describing the bacterial genome have evolved significantly in recent years. Interestingly, probiotic strains of *Lactiplantibacillus plantarum* Tana or *Lactobacillus salivarius* subspecies *salicinius* (SA-03) were isolated from the fecal samples of Olympic athletes and tested in mice for antifatigue effects. In these experiments, probiotic supplementation led to a decrease in lactate, ammonia and creatine kinase [95,96]. Other strains, such as *Lactobacillus plantarum* (TWK10; from pickled vegetables) and *Lactobacillus plantarum* (KSFY01; from yak yogurt), increased glycogen storage, muscle mass and strength, and time to exhaustion in mice [97,98]. In rats, *Bacillus subtilis* (BSB3) was found to prevent negative changes in the gut caused by excessive exercise, and *Saccharomyces boulardii* (Sb) led to aerobic performance enhancement [99,100].

## 8. Concluding Remarks and Future Directions

As gut training becomes more popular among athletes, it is necessary to describe future directions in mapping the interactions between different prebiotics, probiotics, and the most popular ergogenic aids. Very little is known about the modulatory effects of the gut microbiota on the ergogenic actions of most supplements.

While initial meta-analyses on probiotic supplementation in endurance athletes have been published, there is a lack of experiments and meta-analyses in resistance athletes. Most data support the use of single-strain probiotics in aerobic athletes. To date, there have been no experiments on the potential impact of sodium bicarbonate and beta-alanine on the guts of athletes. Although citrulline may have positive effects on anaerobic athletes, the impact of nitrates is less clear. Inulin, FOS, β-glucans, and pectins may play protective roles in gastrointestinal homeostasis, but these effects are not limited to aerobic or anaerobic athletes. Of the prebiotics, only beta-glucans were found to enhance the creatine metabolism pathway and have potential as ergogenic agents; however, data are limited. Sport supplements with potential to modulate that gut microbiota are listed in Table 1. Supplements are described as “Possibly effective” if any human or animal research has suggested increases in health-promoting bacteria abundances and gut function, but these conclusions have not been confirmed in larger samples in double-blind, randomized trials. Table 2 lists products related to gut health that could be studied in athletes in search of ergogenic aid. “Effective” supplements are probiotics that have been found to improve recovery, while the “possibly effective” beta-glucan impact on athletic performance needs to be confirmed in well-designed human experiments.

Considering the immune-boosting effects of probiotics, it can be concluded that ergogenic effects are achieved through a decrease in the number of forced days off (rest days related to infections). Meeting the dietary requirements for fiber (different prebiotic fractions) consumption is a possible way to avoid gastrointestinal tract disturbances during the training and competition season and may protect from potentially harmful nutritional extremes, such as a very high-protein or very high-carbohydrate diet. More studies are needed in the field of both pre- and probiotic supplementation for athletes, as well as ergogenic supplementation of the gut microbiota. Before consuming any nutritional supplement, the person concerned should consult a reliable and qualified professional, who must base his claims on scientific evidence, i.e., a sports doctor or a dietician/nutritionist specializing in sports nutrition.

## Figures and Tables

**Table 1 nutrients-15-01541-t001:** Sport supplements and gut microbiota modulation.

Sport Supplement	Effective	Possibly Effective	Possibly Harmful	Not Known
**Aerobic exercise**				
Maltodextrin/glucose			X	
Sodium bicarbonate				X
Beta-alanine				X
**Anaerobic exercise**				
Protein isolates		X		
Citrulline and arginine		X		
Nitrates				X

**Table 2 nutrients-15-01541-t002:** Gut microbiota modulation and sport performance.

Studied Product	Effective	Possibly Effective	Possibly Harmful	Not Known
**Prebiotics**				
Inulin/FOS				X
Pectin/alginate				X
Beta-glucan		X		
**Probiotics**				
*Lactobacillus* spp.	X			
*Bifidobacterium* spp.	X			

## Data Availability

The data presented in this review are available in MedLine database.

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
