# Peer review of "Different Approaches to Ergogenic, Pre-, and Probiotic Supplementation in Sports with Different Metabolism Characteristics: A Mini Review"

_nutrients, 2023, doi:10.3390/nu15061541_

Round 1

Reviewer 1 Report

Congratulations on the mini-review. You have gathered evidence for the potential benefit of some dietary supplements on different sports activities (anaerobic and aerobic) and their effects on the intestinal microbiota.

A few considerations:

# The article would gain at least one figure and a table. For example, it would be essential to show in a table which nutrients and dietary supplements are genuinely effective and safe (according to the best available evidence) and which are "possibly effective," still without a solid body of evidence.

# In the conclusions, it would be necessary to underline that "before consuming any nutritional supplement, the person concerned should consult a reliable and qualified professional who must base his claims on scientific evidence, i.e., a sports doctor or a dietician-nutritionist specializing in sports nutrition."

  •  
  •  
  •  

Author Response

Thank you for review and your valuable advice:

- two tables (line 373 and 374) and a graphical abstract (line 375) have been added;

- a statement in the conclusions has been added in line 369;

- the manuscript has been extended due to editorial recommendations (additional text describes the results of animal studies) in lines 56-64, 66-67, 90-97, 104-106, 109-111, 182-184, 217-218, 220-223, 231-234, 245, 248-252, 254-263, 278-340.

Reviewer 2 Report

The study is an interesting mini-review that addresses the effects of pre- and probiotics with ergogenic effects on the gut microbiota of athletes. The work concisely summarizes the most current points on the subject (especially the role of citrulline, inulin, FOS, b-glucans, and pectins), highlighting the lack of research on probiotics in sports with anaerobic metabolism.

Authors should correct misspellings in lines 13, 15, 16, 43, 147, 152, 220, 221, 245; use the same abbreviations consistently (e.g. BW vs b.w.; b-glucans vs. beta-glucans; g/kg vs. g / kg), and pay attention to whether p-values and % symbols should be preceded by a white space (or a non-breaking space).

Author Response

Thank you for review and your valuable advice:

- mispellings in lines 13, 15, 16, 43, 147, 152, 220, 221, 245, different abbreviations and white spaces have been corrected;

- the manuscript has been extended due to editorial recommendations (additional text describes the results of animal studies) in lines 56-64, 66-67, 90-97, 104-106, 109-111, 182-184, 217-218, 220-223, 231-234, 245, 248-252, 254-263, 278-340.

- two tables (line 373 and 374) and a graphical abstract (line 375) have been added.